# hnRNP Q/SYNCRIP interacts with LIN28B and modulates the LIN28B/*let-7* axis in human hepatoma cells

**Jason Jei-Sheng Chang**[1,2☯], **Ti Lin**[1☯], **Xin-Yue Jhang**[1], **Shih-Peng Chan**[1,3]*

**1** Graduate Institute of Microbiology, College of Medicine, National Taiwan University, Taipei, Taiwan, **2** Institute of Biomedical Sciences, Academia Sinica, Taipei, Taiwan, **3** Genome and Systems Biology Degree Program, College of Life Science, National Taiwan University, Taipei, Taiwan

☯ These authors contributed equally to this work.

* shihpengchan@ntu.edu.tw

**Data Availability Statement:** All relevant data has been uploaded to the OSF repository and can be accessed at this link: https://doi.org/10.17605/OSF.IO/JR2U8.

## Abstract

The RNA-binding protein LIN28B represses the biogenesis of the tumor suppressor *let-7*. The LIN28B/*let-7* axis regulates cell differentiation and is associated with various cancers. The RNA-binding protein Q (hnRNP Q) or SYNCRIP (Synaptotagmin Binding Cytoplasmic RNA Interacting Protein) has been implicated in mRNA splicing, mRNA transport, translation, and miRNAs biogenesis as well as metabolism in cancer. To determine whether hnRNP Q plays a role in the LIN28B/*let-7* axis, we tested for interactions between hnRNP Q and LIN28B. We demonstrated that hnRNP Q interacts with LIN28B in an RNA-dependent manner. Knockdown of hnRNP Q caused reduced expression of a well-known *let-7* target TRIM71, an E3 ubiquitin ligase that belongs to the RBCC/TRIM family, and also LIN28B, whose mRNA itself is down-regulated by *let-7*. In addition, hnRNP Q knockdown increased *let-7* family miRNA levels and reduced the activity of luciferase reporters fused with the *TRIM71* 3'UTR or a synthetic 3'UTR carrying 8X *let-7* complementary sites. Finally, depletion of hnRNP Q inhibited the proliferation of a hepatocellular carcinoma cell line, Huh7. This observation is consistent with the survival curve for liver cancer patients from the TCGA database, which indicates that high expression of hnRNP Q is a prognostic marker for a poor outcome in individuals afflicted with hepatocellular carcinoma. Together, our findings suggest that hnRNP Q interacts with LIN28B and modulates the LIN28B/*let-7* axis in hepatocellular carcinoma.

## Introduction

Post-transcriptional regulation is extensively modulated by RNA-binding proteins (RBPs). Previous screens have identified over 1,500 RBPs that account for ~7.5% of all protein-coding genes in humans [1]. RBPs interact with different types of RNAs, including mRNAs, miRNAs, snRNAs, snoRNAs, and lncRNAs, and regulate their processing, function, alternative splicing, polyadenylation, localization, stability, and translation [2]. Most RBPs contain at least one RNA-binding domain (RBD), a category that includes RNA-recognition motifs (RRM), K-

**Funding:** 1.Ministry of Science and Technology, Taiwan (MOST 105-2311-B-002-011-MY3) 2. Ministry of Science and Technology, Taiwan (MOST 108-2311-B-002-012) 3.Ministry of Science and Technology, Taiwan (MOST 111-2314-B002-065-MY3) 4.National Taiwan University Hospital (UN108-040) 5.National Taiwan University (109L7224) 6.National Taiwan University (110L7205) 7.National Taiwan University Hospital Yunlin Branch (NTUHYL110.I003) 8.National Taiwan University Hospital Yunlin Branch (NTUHYL111.I009) The funders had no role in study design, data collection and analysis, decision to publish, or preparation of the manuscript.

**Competing interests:** The authors have declared that no competing interests exist.

homology (KH) domains, zinc-finger domains, and arginine/glycine-rich (RGG/RG) domains; RBDs can have sequence-dependent or structure-specific interactions with target RNAs [3]. In human cancers, aberrant levels of RBPs have been shown to affect target RNAs that are linked to proliferation, apoptosis, angiogenesis, senescence of cancer cells, and epithelial-mesenchymal transition (EMT)/invasion/metastasis [4]. Several types of RBPs have been demonstrated to regulate certain oncogenic or tumor-suppressive miRNAs. Abnormalities of these RBPs participate in carcinogenesis and progression of various cancers, as well as genetic diseases, making them important prognostic markers and therapeutic targets [5–7].

Many RBPs regulate miRNA biogenesis and function [5, 8]. Among them, LIN28 selectively inhibits the maturation of the *let-7* miRNA family by bipartite interactions with terminal-loop structures of the precursor miRNA through LIN28's cold-shock and zinc-knuckle domains (CSD and ZKD, respectively) [9, 10]. A single *lin-28* gene represses *let-7* expression in the heterochronic gene regulatory pathway that governs larval development in *C. elegans* [11, 12]. The mammalian genome contains two LIN28 paralogs, LIN28A and LIN28B [13, 14]. Although neither LIN28A nor LIN28B are usually present in mature tissues, they are re-expressed in several cancers to support cancer growth [15–17]. Interestingly, the expression of LIN28A and LIN28B is mutually exclusive in many human cancer cell lines [18]. In the cytoplasm, the binding of LIN28A recruits the terminal uridylyltransferase (TUTase), Zcchc11, to the precursor *let-7* molecules and leads to oligouridylation followed by rapid degradation of *let-7* [19, 20]. LIN28B was found to localize in the nucleus and nucleolus, where its binding to the primary *let-7* molecules inhibits processing by the Microprocessor [21–23]. Some conflicting studies, however, have shown that LIN28B is predominantly cytoplasmic, suggesting that the precise subcellular location of human LIN28B might vary in different cell types or LIN28B may shuttle into the nucleus in a cell cycle-dependent manner [13, 24].

Expression of *let-7* is required to repress downstream genes including *LIN-41/TRIM71*, which encodes an RBP of the TRIM/RBCC family required for maintenance of stemness [25]. The repression of *let-7* biogenesis by LIN28 functions as a pluripotency factor and possibly a tumorigenesis driver [18]. Since the expression of LIN28 is itself downregulated by *let-7* miRNAs, LIN28 and *let-7* comprise a bi-molecular switch to control cell differentiation [26]. Interestingly, recent studies have shown that TRIM71 represses the maturation of *let-7* miRNAs by binding to and functionally cooperating with LIN28 [27]. TRIM71 also represses Ago2 mRNA translation, resulting in lower Ago2 availability and corresponding changes in levels of *let-7* [28]. This reciprocal down-regulation of *let-7* and TRIM71 serves as an additional downstream switch. This LIN28/*let-7* axis is demonstrated as evolutionarily conserved from nematode to human [29]. Dysfunction of this axis may turn normal cells into highly aggressive tumors with elevated LIN28 levels.

In addition to regulating *let-7* biogenesis, LIN28 regulates the translation of many mRNAs in a *let-7*-independent manner [24, 30–34]. Many studies using CLIP or PAR-CLIP have revealed a broad range of LIN28 target mRNAs and the molecular requirement for LIN28 binding [24, 35–37]. However, there are relatively few studies examining whether or how LIN28 cooperates with other RBPs on mRNAs or miRNAs. Recently, a study in mouse embryonic stem cells (ESCs) employed a proteomic procedure to identify possible Lin28A partners in Lin28A-containing high-molecular-weight complexes and found numerous protein candidates. There has been evidence that the suppression of Ddx3x, Hnrnph1, Hnrnpu, and Syncrip (Hnrnpq) interferes with the interaction of Lin28A with Dnmt3a mRNA [38].

We are interested in the possible interaction between hnRNP Q (the synaptotagmin-binding cytoplasmic RNA interacting protein [SYNCRIP]) and LIN28 because abnormal levels of hnRNP Q have been implicated in several types of cancers, such as leukemia [39], liver cancer [40], and colorectal cancer [41]. Here, we demonstrate that human hnRNP Q interacts with

LIN28B by co-immunoprecipitation in hepatocellular carcinoma cells. Suppression of hnRNP Q causes an increase in the *let-7* miRNA levels and affects the regulation of the 3′UTRs bearing the *let-7* complementary sequences, including the 3′UTR of the oncogenic *TRIM71* gene. These results suggest that by association with LIN28B-containing RNP complexes, hnRNP Q plays a role in *let-7* biogenesis. Alternatively, by binding to the same RNA targets of LIN28B, hnRNP Q could potentially modulate regulation by LIN28B, which in turn might influence levels of *let-7* and the downstream expression of TRIM71.

## Materials and methods

### Cell culture and RNA interference

Huh7, HEK293, and HEK293T cell lines were cultured in Dulbecco's modified Eagle's medium (DMEM; Gibco, 11965–084) supplemented with 2 mM L-glutamine (Corning, 25-005-CI), 100 U/ml penicillin/streptomycin mixture (Corning, 30-004-CI), and 10% heat-inactivated fetal bovine serum (FBS; Gibco, 10437–028) at 37°C and in a humidified atmosphere containing 5% $CO_2$. The Huh7 cell line is a gift from Dr. James Ou's lab at the University of California.

To perform RNAi, siRNAs targeting hnRNP Q were obtained from Invitrogen (si_HSS116100/ si_hnRNP Q#1, 5' GGACCACCTCCAGATTCCGTTTATT 3'; si_HSS193680/ si_hnRNP Q#2, 5' GACCTATATGGGATCTTCGTCTAAT 3'; and si_HSS116098/ si_hnRNP Q#3, 5' GCTCAGGAGGCTGTTAAACTGTATA 3'). For siRNA transfection in Huh7 cells, the complete medium was replaced with free-DMEM (DMEM without L-glutamine, antibiotics and FBS) one hour before transfection. To knock down hnRNP Q, two rounds of silencing were performed. For the first round, $3×10^5$ suspended cells were cultured in free-DMEM containing 100 nM siRNA in a 6-well plate for 24 hr. The second round was on the next day when cells were transfected again with 100 nM siRNA. After two rounds of transfection, cells were maintained for 24 hr in the culture medium before lysis.

For lentivirus production, HEK293T cells were co-transfected with 1 μg pMD2.G 1, 9 μg pCMVΔR8.91, and 10 μg pLKO plasmids encoding specific shRNA sequences (Academia Sinica, Taipei, Taiwan). $1.2×10^6$ cells were seeded a day before transfection. After transfecting for 24 hr, the free-DMEM containing transfection reagent was removed and replaced with culture medium (completed medium). The supernatant containing lentivirus was collected and filtered through a 0.45 μm filter (Sartorius Stedim Biotech, 16555). To conduct virus infection, $1.2×10^6$ cells were seeded on a 10 cm dish a day before transduction. 1 ml virus-containing supernatant was added to the culture medium supplemented with 6.4 μg/ml polybrene (Sigma, 107689). Cells were maintained for 24 hr after transduction and then selected with 2 μg/ml puromycin (Sigma, P8833) for 5–7 additional days. Two clones obtained from Academia Sinica were used to knock down hnRNP Q expression in HEK293 and Huh7 cells (TRCN0000275206/ sh_hnRNP Q#1, 5'-GTATGACGATTACTACTATTA-3'; TRCN0000275205/ sh_hnRNP Q#2, 5'-TATATGGGATCTTCGTCTAAT-3'; and TRCN0000231722/shLacZ, 5'-CGCGATCGTAATCACCCGAGT-3' as control).

### Plasmid construction

RNAi-resistant hnRNP Q expression vectors were composed of hnRNP Q1, Q2, or Q3 coding sequences, respectively, with three HA tags fused at the C-terminus. They were cloned from pEF-FLAG plasmids carrying wild-type hnRNP Q1, Q2 or Q3 (gifts from Dr. Woan-Yuh Tarn, Academia Sinica, Taipei, Taiwan) [42]. The HA sequences from pKH3 (Addgene, #12555) were PCR amplified and used to replace the FLAG tag of the pEF-FLAG backbone using the HA Fwd (containing a *Not*I site) forward primer, and HA Rev (containing a *Bst*BI

site) reverse primer to generate pEF-3HA. To generate RNAi-resistant pEF-hnRNP Q1/2/3-3HA, two primer pairs hnRNP Qr#1 Fwd1/Rev1 and hnRNP Qr#1 Fwd2/Rev2 were used to produce two overlapping hnRNP Q1/2/3 fragments with nucleotide replacements designed in the overlapping region. These two fragments were assembled using NEBuilder and amplified by PCR using hnRNP Qr#1 Fwd1 and Rev2 before being cloned into pEF-3HA by *Not*I.

To construct psiCHECK2-*TRIM71* 3' UTR 468 bp, bases 1 to 468 of the *TRIM71* 3' UTR was custom synthesized and cloned into psiCHECK2 at the *Xho*I and *Not*I restriction sites. For psiCHECK2-*TRIM71* 3' UTR 424 bp, two *let-7* complementary sequences (LCSs) of the *TRIM71* 3' UTR were designed to be removed when synthesized. As for psiCHECK2-*TRIM71* 3' UTR 6,015 bp, five sets of primers were designed according to DNA sequences from NCBI (*TRIM71* 3' UTR Fwd1-5/ Rev1-5; 6,015 bp in length). 3' UTR fragments amplified from Huh7 genomic DNA were ligated with *Not*I-linearized psiCHECK2-*TRIM71* 3' UTR 468 bp by NEBuilder at 50˚C for an hour. The *TRIM71* 3' UTRs were also cloned to another dual luciferase plasmid, pmirGLO by using the *Xho*I and *Not*I restriction sites. psiCHECK2 plasmids containing eight tandem repeats of *let-7* target or mutant sequences were obtained from Addgene (#20931) [43]. All primer sequences used for cloning are listed in **S1 Table**.

## Immunoprecipitation

For immunoprecipitation, cells were lysed in NET-2 buffer (50 mM Tris-HCl pH7.5, 150 mM NaCl, 0.05% Triton-X100) containing 1X Protease Inhibitor Cocktail (Roche, 4693132001). 1 μg antibody against hnRNP Q/SYNCRIP (Atlas Antibodies, HPA041275), LIN28B (Cell Signaling, #4196), or rabbit IgG1 (Sigma, I5006) was conjugated to 50 μl Dynabeads M-280 sheep anti-rabbit IgG (Invitrogen, 11204D) at room temperature for one hour and washed with NET-2 buffer containing 1X protease inhibitor three times. For HA immunoprecipitation, 1 μg rat anti-HA (Roche, 11867423001), rat IgG (Sigma, I4131), and Dynabeads Protein G (Invitrogen, 10003D) were used. As for protein binding, 1,000 μg to 2,500 μg of protein was incubated with antibody-conjugated Dynabeads at 4˚C for 2 hr. After collecting the supernatant and washing three times with NET-2 containing protease inhibitor, Dynabeads were separated into two parts. One was treated with 50 μg RNase A at room temperature for 20 min while the other was untreated. Lastly, the RNase-treated Dynabeads were washed a further three times with NET-2 buffer containing protease inhibitor.

## Immunoblotting

Protein samples were separated by SDS-PAGE and transferred to PVDF membranes. The following antibodies were used for protein detection: TRIM71 (R&D system, AF5104, 1:3,000), hnRNP Q/R (Sigma, R5653, 1:5,000), LIN28B (Cell Signaling, #4196, 1:5,000), HA (Covance, MMS101R, 1:5,000), β ACTIN (Protein Tech, 66009-1-Ig, 1:10,000), anti-sheep (Jackson ImmunoResearch, 713-035-003, 1:5,000), anti-mouse (Sigma, A9044, 1:10,000), and anti-rabbit (Jackson ImmunoResearch, 211-032-171, 1:5,000). WesternBright ECL HRP substrate (Advansta K-12045-D20) or WesternBright Sirius HRP substrate (for detecting TRIM71 only) (Advansta K-12045-D10) were used for signal detection by the BioSpectrum AC Imaging System (UVP). Images were analyzed and quantified by the ImageLab Software (BIO-RAD).

## Immunofluorescence assay (IFA)

The localization of HA-tagged hnRNP Qs was determined through an immunofluorescence assay (IFA). $1 \times 10^5$ cells were seeded a day before transfection in a 12-well plate with a cover slip. After transfecting hnRNP Q plasmids for 24 hr, cells were fixed with 4% paraformaldehyde diluted in PBS for 20 min and permeabilized by 1% Triton X-100 diluted in PBS at room

temperature for 5 min. To inactivate free aldehydes that might cause background signal, the coverslip was incubated in 1% BSA dissolved in PBST (PBS with 0.1% Tween 20) containing 0.3 M glycine at room temperature for 30 min. Before hybridizing with the primary antibody, the coverslip was further incubated with 1% BSA/PBST at room temperature for another 1.5 hr.

For primary antibody hybridization, HA (Covance, MMS101R, 1:1,000), hnRNP Q/R (Sigma, R5653, 1:5,000), or LIN28B (Cell Signaling, #4196, 1:5,000) was dissolved in 1% BSA/PBST and incubated with the coverslip at 4˚C overnight. After washing three times with PBST, the coverslip was further incubated with Goat Anti-Mouse IgG H&L (Alexa Fluor 488; Abcam, ab150113, 1:1,000) or Cy3 AffiniPure Goat Anti-Rabbit IgG (H+L; Jackson ImmunoResearch, 111-165-144) at 37˚C for one hour followed by three additional washes, DAPI staining, and mounting.

## Cytoplasmic/nuclear fractionation

To separate proteins from the cytoplasm and nucleus, nuclei isolation buffer (10 mM HEPES-KOH pH 7.9, 10 mM KCl, 1.5 mM $MgCl_2$, 10% glycerol, 0.34 M sucrose, 1 mM DTT, 0.1% Triton X-100) with 1X Protease Inhibitor Cocktail (Roche, 4693132001) was used to lyse cells. After a 10 min incubation on ice, cells were centrifuged at $15,000 \times g$ for one min at 4˚C and collected in a new Eppendorf tube. The cell nuclei were further lysed by RIPA with protease inhibitor on ice for 20 min and centrifuged at $15,000 \times g$ for 5 min at 4˚C followed by collection.

## Northern blotting

Northern blotting optimized for small RNA detection was used to detect the *let-7* miRNAs in hnRNP Q depleted cells. Briefly, 20 μg RNA extracted from cells was mixed with 2× loading dye (95% formamide, 2.5% BPB, and 2.5% XC), heated at 95˚C for 5 min, and separated by electrophoresis in 0.5× TBE using a 12% acrylamide gel with 8M urea which was pre-run at 350 V for one hour. Separated RNA was transferred to a Hybond $N^+$ membrane (GE healthcare, #RPN119B) in 0.5× TBE at 10 V overnight. After transfer, the membrane was air-dried, UV crosslinked using a Spectrolinker XL-1500 UV crosslinker (Spectronics) on both sides, and baked at 80˚C for one hour followed by pre-hybridization in pre-hybridization buffer (4× SSPE and 7% SDS, hereafter pre-hy buffer) at 42˚C for one hour.

To prepare probes used in small RNA northern blotting, 10 pmol anti-*let-7-a* (`AAC TAT ACA ACC TAC TAC CTC A`) and anti-U6 (`TCA CGA ATT TGC GTG TCA TCC T`) DNA oligonucleotide were labeled with γ-32P (PerkinElmer, BLU502) by polynucleotide kinase (NEB, M0201). After filtering with the MicroSpin G-25 column (GE Healthcare, 27-5325-01), the probe was dissolved in pre-hy buffer and hybridized with the membrane at 42˚C overnight. The membrane was washed twice with low stringency buffer (4 × SSPE and 4% SDS) at 42˚C for 15 min and high stringency buffer (0.1× SSC and 0.5% SDS) at room temperature for 5 min. A phosphor image plate (Fujifilm, BAS-IP MS 2025) and Typhoon 5 Trio Variable Mode Imager (GE Healthcare) were used to detect and quantify the irradiation signal.

## TaqMan quantitative PCR (qPCR)

TaqMan qPCR was used to measure the changes in expression of members of the *let-7* miRNA family after hnRNP Q depletion. 100 ng of total RNA was subjected to reverse transcription using TaqMan stem-loop RT primers (Applied Biosystems) for specific *let-7* family miRNAs and MultiScribe Reverse Transcriptase (Applied Biosystems, 4311235). TaqMan MicroRNA Assay 20× (Applied Biosystems) containing miRNA isotype-specific primers and MGB probes

was combined with TaqMan 2× Universal Master Mix II (Applied Biosystems, 4440043) to perform the quantitative PCR with a StepOnePlus Real-Time PCR System (Applied Biosystems, 4376600) following the manufacturer's instructions. The expression profile for each *let-7* family miRNA was analyzed using the ΔΔCt method, and the expression levels were calibrated using U6 or SNU48 as internal controls.

## Luciferase assay

For the HEK293 luciferase assay, $4.5 \times 10^5$ cells were seeded in a 6-well plate a day before transfection. Next, 1 μg of dual luciferase plasmid was delivered into hnRNP Q depleted HEK293 cells using Lipofectamine 2000 (Invitrogen, 116680). After 24 hr, cells were harvested and lysed with RIPA (150 mM NaCl, 5 mM EDTA pH 8.0, 1 M Tris-base pH 8.0, 1% NP-40, 0.5% sodium deoxycholate, and 0.1% SDS) containing 1X Protease Inhibitor Cocktail (Roche, 4693132001). For the Huh7 luciferase assay, $3 \times 10^5$ cells were seeded in a 6-well plate a day before transfection. After delivering 1 μg dual luciferase plasmid into hnRNP Q-depleted Huh7 cells using the HyFect DNA transfection reagent (LEADGENE, 30201), cells were maintained in culture medium (completed medium) for 24 hr and harvested (48 hr post-transfection). A dual-luciferase reporter assay system (Promega, E1910) was used to measure the relative luciferase activities according to the manufacturer's instructions. Signals were obtained using a Berthold Orion II Microplate Lumimometer or Spectra Max i3x Multi-Mode Microplate Reader.

## MTT assay

Huh7 cell lines were first cultured in a 10 cm dish and infected with lentivirus carrying sh_Lac Z, sh_hnRNP Q#1, or sh_hnRNP Q#2 plasmid. After infected cells were selected using 10 mM puromycin for 3 cell passages, they were re-seeded in a 96-well microplate at a density of 3000 cells per well and incubated in DMEM + Glutamine for 4 days with medium changed every two days. Cell proliferation was measured by 3-(4,5-dimethylthiazol-2-yl)-2,5-diphenyl tetrazolium bromide (MTT) (EMD Biosciences, San Diego, CA, USA) assays on day 0 (14hr), 1 (38hr), 2 (62hr), and 3 (86hr) (n = 3 replicates). Each well received a 100 μL mixture consisting of 10 μL of MTT solution with 90 μL of RPMI medium and was then incubated for 2 h. The absorbance was measured at 570 nm.

## Survival analysis

The clinical data for hnRNP Q/SYNCRIP expression levels correlating with survival was downloaded from the Human Protein Atlas and Genomic Data Commons Portal (GDC). RNAseq samples from 365 patients with liver cancer were examined. FPKM (fragments per kilobase of transcript per million mapped reads) values for hnRNP Q/SYNCRIP greater than 12.43 was defined as high expression according to the Human Protein Atlas. The Mantel-Cox test was used to analyze the data sets.

## Statistical analysis

The data are illustrated as mean ± SEM and statistical analyses were performed using Graph-Pad, Prism 9 (San Diego, CA). Unpaired two-tailed t-tests was used to compare means.

# Results

## hnRNP Q/SYNCRIP interacts with LIN28B in an RNA-dependent manner

The hnRNP Q family contains three major isoforms (Q1, Q2, Q3) resulting from alternative splicing [42]. The most abundant hnRNP Q1 is characterized by an acidic residue-rich domain

at its N terminus, three RNA recognition motifs (RRMs) in the middle, and an arginine/glycine (RG)-rich domain at its C terminus. The Q2 isoform contains an extended C-terminal region but a truncated RRM2. The longest isoform Q3 contains the same three RRMs as Q1 and the same extended C-terminal as Q2. With respect to endogenous localization, only hnRNP Q1 shuttles between the cytoplasm and nucleus, while hnRNP Q2 and Q3 predominantly localize to the nucleus [44].

We wanted to test if hnRNP Q interacts with LIN28 in human hepatocarcinoma Huh7 cells, which express a high level of LIN28B [13] but not LIN28A. First, we conducted immunoprecipitation using an antibody against endogenous hnRNP Q proteins in Huh7 lysates. Interestingly, we detected a significant amount of LIN28B in the precipitates, indicating an interaction between these two RNA binding proteins (**Fig 1A**). Treatment of RNase A on the pellet largely reduced the signal, suggesting an RNA-dependent interaction. A reciprocal co-immunoprecipitation using an antibody against endogenous LIN28B also pulled down hnRNP Q in an RNA-dependent manner (**Fig 1B**). To further validate the interaction with hnRNP Q and LIN28B, we cloned the three major hnRNP Q isoforms, Q1, Q2, and Q3, with sequences resistant to sh_*hnRNP Q*#1, into the pEF-3HA plasmid (**S1 Fig**). We expressedthese HA-tagged hnRNP Q isoforms (Q1, Q2, Q3) separately into Huh7 cells to establish three lines. For each line we induced expression of the HA-tagged hnRNP Q isoforms, collected cell lysates, and then conducted anti-HA immunoprecipitation on the cell lysates. We detected RNA-dependent interactions between all three major HA-tagged hnRNP Q isoforms and LIN28B (**Fig 1C–1E**), indicating that despite differences in structure and localization, the hnRNP Q isoforms may share RNA targets with LIN28 or precipitate into the same ribonucleotide protein particles within LIN28B.

## hnRNP Q/SYNCRIP depletion reduces protein levels of *let-7* targets TRIM71 and LIN28B in Huh7 cells

To investigate whether hnRNP Q could play a role in the regulation of the LIN28/*let-7* axis, we first sought to determine whether protein levels of LIN28B and another well-known *let-7* target, LIN41/TRIM71, were affected by depleting hnRNP Q through RNAi using lentivirus-driven shRNAs. Results from two stably selected sh_hnRNP Q clones showed effective depletion of hnRNP Q, as well as associated decreases in both LIN28B and TRIM71 (**Fig 2A–2D and S2 Fig**). A similar pattern was seen when we used exogenous siRNAs to reduce hnRNP Q expression (**S3A-S3D Fig**), although decreases in LIN28B and TRIM71 levels were more subtle. One possible explanation for the smaller degree of decrease in the siRNA experiments could be due to differences in the assays; the shRNA experiments used stable clones while the siRNA transfection experiments did not.

Next, we wanted to test whether the ectopic expression of RNAi-resistant hnRNP Q proteins could restore the phenotypes seen in hnRNP Q knockdown cells. RNAi-resistant HA-hnRNP Qs were transfected into Huh7 cells with hnRNP Q depleted by sh_*hnRNP Q*#1. We found that all of the RNAi-resistant hnRNP Q isoforms exhibited the same localization patterns as endogenous hnRNP isoforms; hnRNP Q1 was located in both the nucleus and cytoplasm, whereas Q2 and Q3 were exclusively in the nucleus (**S4 Fig**). We found that adding back RNAi-resistant hnRNP Qs increased protein levels of both TRIM71 and LIN28B (**Fig 2E**), supporting the possibility that changing the amount of hnRNP Q isoforms in cells could affect their interaction with LIN28 and the downstream target TRIM71.

## Depletion of hnRNP Q increases *let-7* miRNA levels

We next sought to determine whether hnRNP Q depletion affects *let-7* miRNA levels. We conducted northern blotting to investigate whether downregulation of LIN28B caused by hnRNP

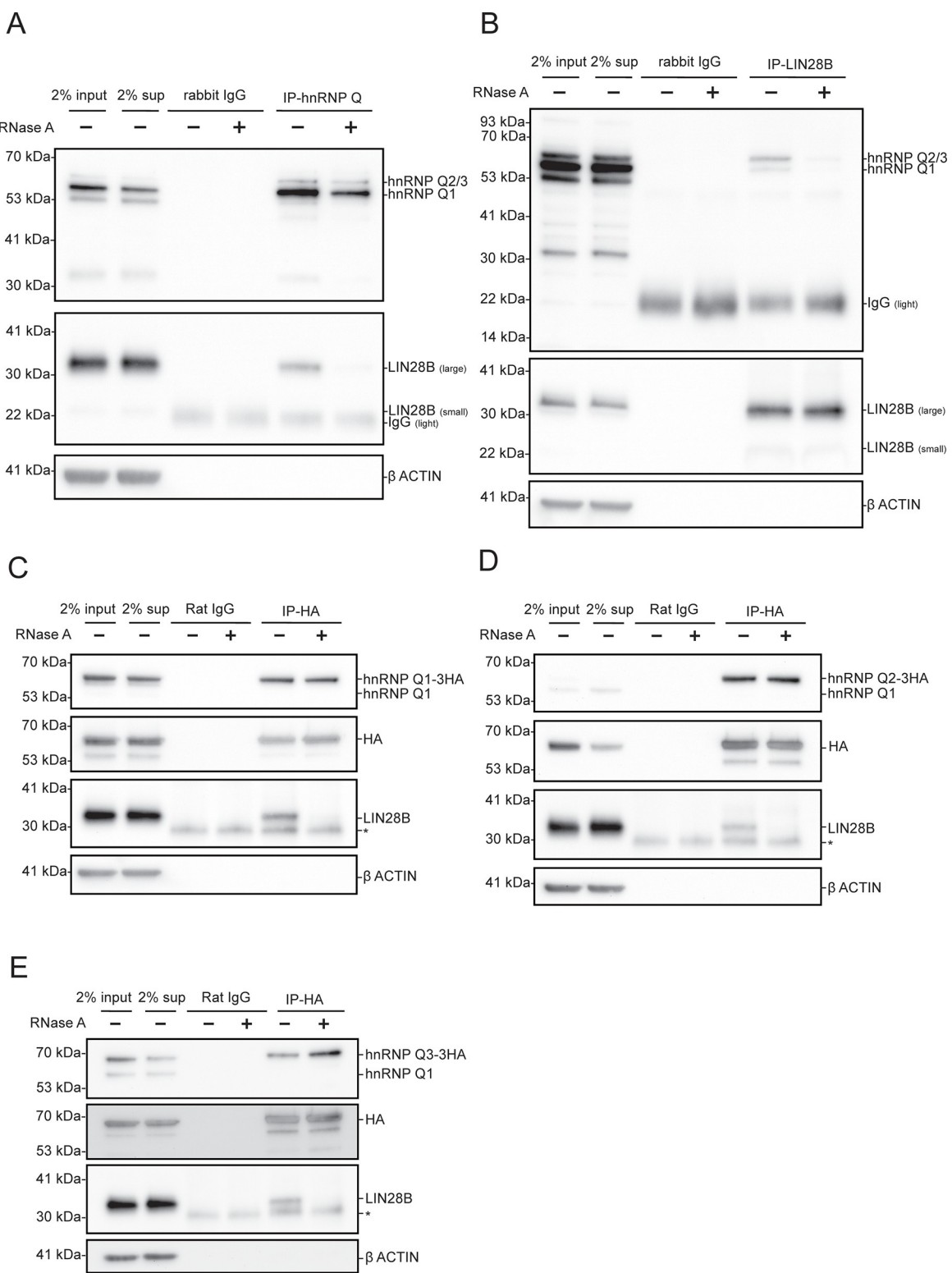

**Fig 1. Reciprocal immunoprecipitation showed that hnRNP Q interacts with LIN28B in an RNA-dependent manner in Huh7 cells.** Immunoblots showing the RNA-dependent coprecipitation of (**A**) endogenous LIN28B with endogenous hnRNP Q, (**B**) endogenous hnRNP Qs with endogenous LIN28B, (**C**) endogenous LIN28B with ectopic hnRNP Q1-3HA, (**D**) endogenous LIN28B with ectopic hnRNP Q2-3HA, and (**E**) endogenous LIN28B with ectopic hnRNP Q3-3HA in Huh7 cells. *Represents an unknown non-specific band.

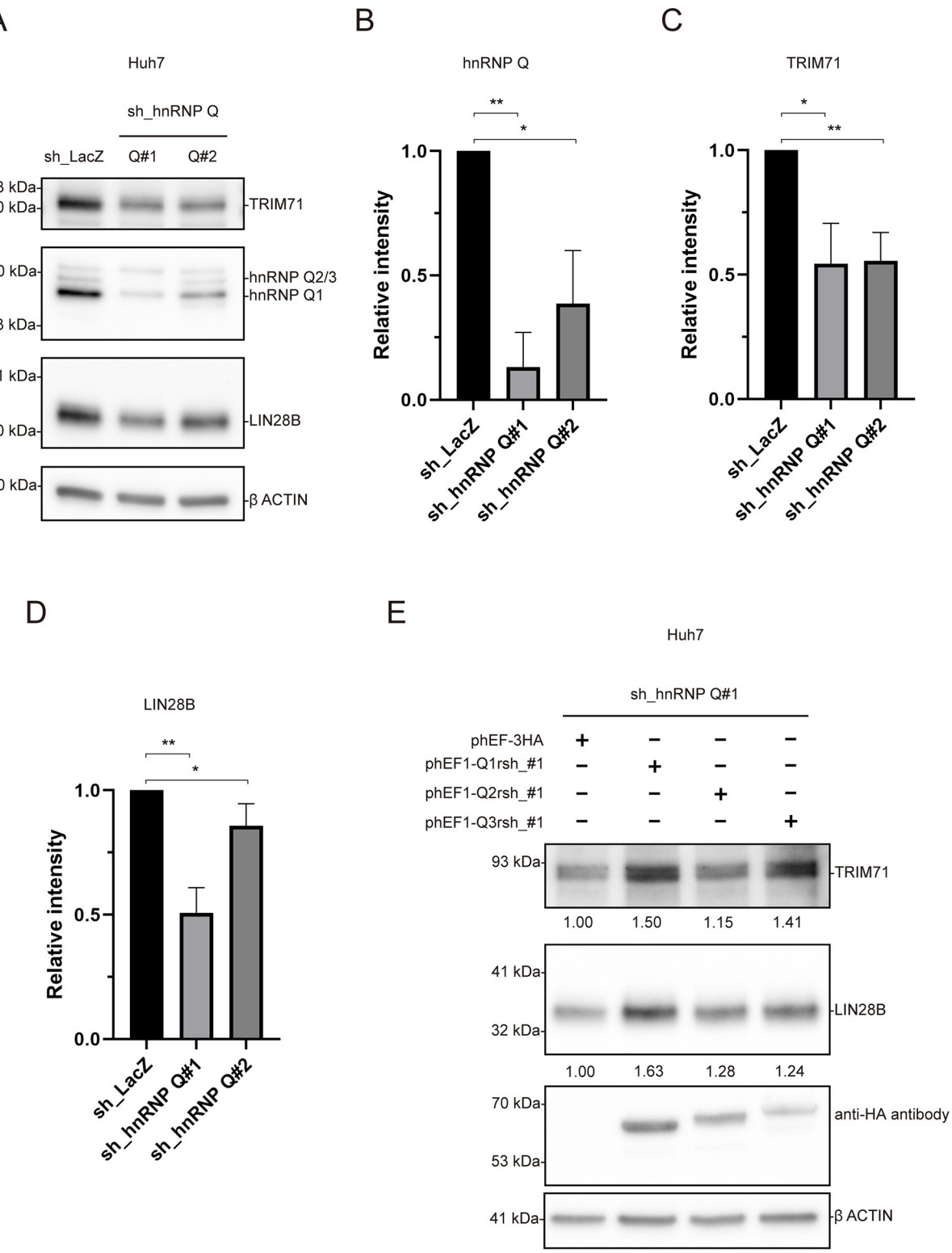

**Fig 2. Effectively depleting hnRNP Q in Huh7 cells reduced TRIM71 and LIN28B protein expression levels. (A)** Depleting hnRNP Q by sh_hnRNP Q reduced TRIM71 and LIN28B in Huh7 cells. Two different shRNAs (Q#1, TRCN0000275206 and Q#2, TRCN0000275205) were used. **(B)** Quantification of hnRNP Q levels in Huh7 cells by shRNA. **(C)** Quantification of TRIM71 levels in hnRNP Q depleted Huh7 cells. **(D)** Quantification of LIN28B levels in hnRNP Q depleted Huh7 cells. **(E)** Re-expression of ectopic RNAi-resistant hnRNP Q1/2/3-3HA increased the protein levels of TRIM71 and LIN28B. One representative western blot is shown here. The other replicates are shown in **S2 Fig**. Results are plotted as average ± S.D., *$P < 0.05$, **$P < 0.01$, ***$P < 0.001$ using an unpaired two-tailed Student's *t*-test.

Q depletion was accompanied by a change in *let-7* levels. A $^{32}$P 5′ end-labeled oligonucleotide with sequences complementary to human *let-7a* was used as the probe. Our results showed that hnRNP Q depletion increased *let-7-a* miRNA levels in Huh7 cells (**Fig 3A, 3B** **and S5A Fig**). We also knocked-down hnRNP Q in HEK293 cells and observed similar results (**Fig 3C, 3D** **and S5B Fig**). By contrast, hnRNP Q depletion did not influence levels of miR-21, a control microRNA not found in the LIN28/*let-7* axis, in Huh7 cells (**Fig 3D, 3E** **and S5C Fig**). We hypothesize that hnRNP Q may contribute to LIN-28's inhibition of pri- or pre-*let-7* processing, since hnRNP Q has been reported to interact with LIN28B as well as the terminal loop of pri-*let-7a* [45]. However, hnRNP Q was previously described as a repressor of pri-*let-7a* processing.

Next, we used TaqMan qRT-PCR to examine the relative expression of *let-7* miRNA isoforms in Huh7 cells upon hnRNP Q knockdown. Examination of *let-7* levels showed an increase in each *let-7* isoform (**Fig 4A and S6 Fig**). To support the idea that decreases in TRIM71 protein levels were due to elevated *let-7* levels caused by hnRNP Q depletion, we transfected hnRNP Q-depleted cells with *let-7c* inhibitor (anti-*let-7c*) and found the protein level of TRIM71 increased (**Fig 4B**).

## hnRNP Q depletion affects the regulation of the *let-7* targeted TRIM71 3′UTR and the proliferation of Huh7

To determine whether the decrease of TRIM71 upon hnRNP Q knockdown was due to elevated *let-7* levels, we cloned a 468-bp fragment of the TRIM71 3′UTR sequence (RefSeq NM_001039111.1) containing two *let-7* complementary sequences (LCSs) into the psiCHECK2 vector and performed luciferase activity assays. A 424-bp TRIM71 3′UTRΔLCS with deletions for those two *let-7* complementary sequences was used for comparison (**S7A and S7B Fig**). sh_hnRNP Q#1 was chosen for the knockdown assays because it exhibited the highest knockdown efficiency. In HEK293 cells, depletion of hnRNP Q was associated with a significant reduction in the activity of Renilla luciferase whose transcripts carried the TRIM71 3′UTR. By contrast, the psiCHECK2 reporter fused to TRIM71 3′UTRΔLCS showed no change in Renilla luciferase activity following a comparable reduction in hnRNP Q levels (**Fig 5A**). This suggests that hnRNP Q knockdown elevated *let-7* miRNA levels which then down-regulates TRIM71 expression through the 3′UTR. To support this, we also employed two other psiCHECK2 reporters for luciferase activity assays, one carrying eight copies of the LCS in the 3′UTR and another carrying eight mutated sites (**S7C Fig**) [43]. Again, depletion of hnRNP Q only reduced the Renilla luciferase activity of the vector with normal LCSs (**Fig 5B**).

To further support this notion, we cloned the above 468-bp or full-length 6015-bp TRIM71 3′UTRs (RefSeq NM_001039111.2), which carries seven LCSs, into another dual-luciferase vector pmirGLO. The pmirGLO vector harbors the human phosphoglycerate promoter to drive luciferase at the same transcriptional efficiency as housekeeping genes (**S7D and S7E Fig**). As seen in **Fig 5C**, depletion of hnRNP Q significantly reduced the expression of the luciferase reporter ligated to both predicted versions of the TRIM71 3′UTR in Huh7 cells. TRIM71 has been implicated in the growth and tumorigenicity of hepatocellular carcinoma [46], and LIN28B has been demonstrated to promote hepatocellular carcinoma cell progression as well [47]. Also, an MTT assay showed that knocking down hnRNP Q would notably inhibit Huh7 proliferation (**Fig 5D**). Thus, we infer that depletion of the hnRNP Q protein could lead to an increase of *let-7* family miRNAs, which in turn would inhibit expression of their targets (LIN28B and TRIM71), and thereby repress hepatocellular carcinogenesis. Our results are consistent with the survival curve for hepatocellular carcinoma patients from the

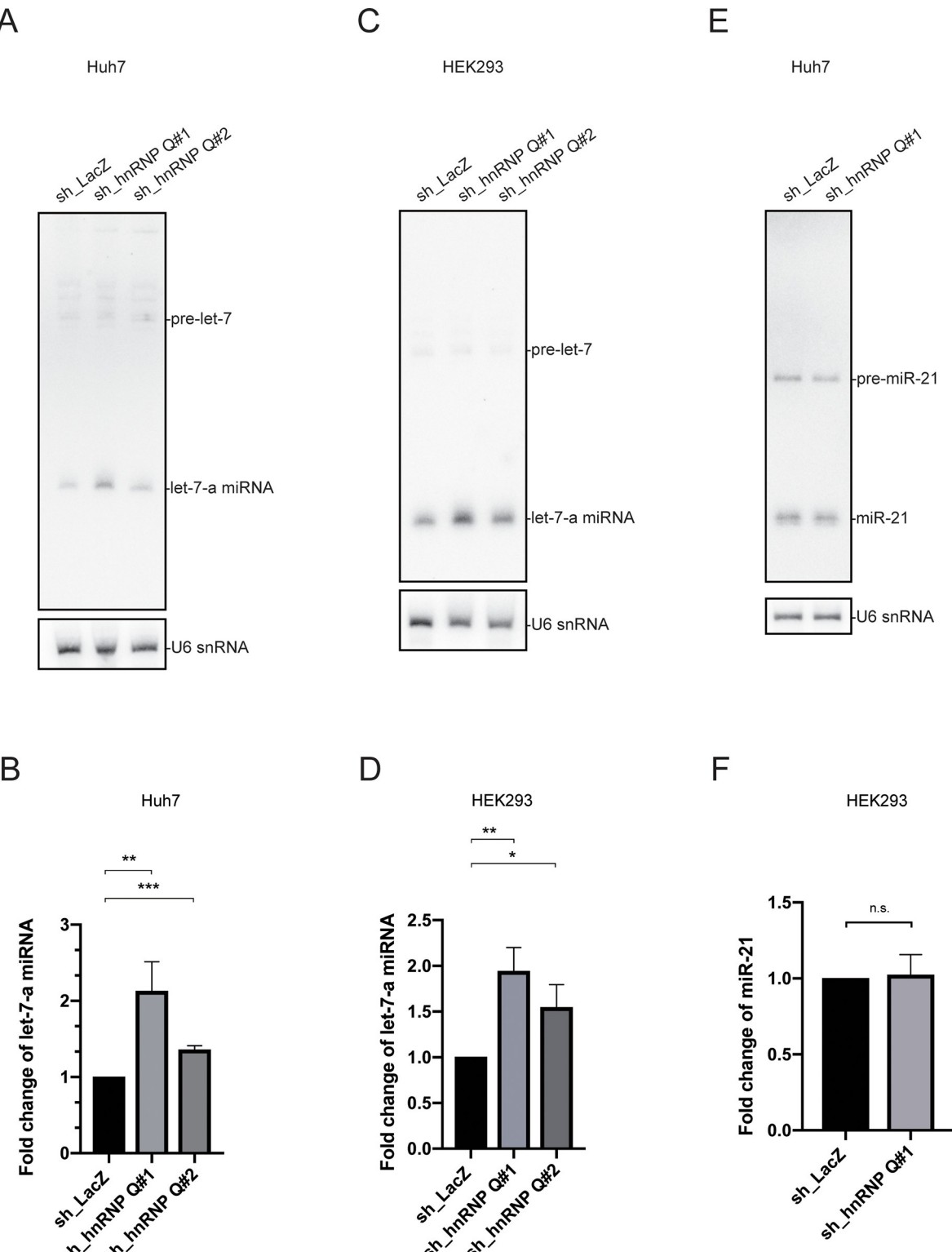

**Fig 3. hnRNP Q depletion increased *let-7* miRNA levels in Huh7 cells or HEK293 cells. (A)** Northern blot showing increased *let-7-a* levels upon depletion of hnRNP Q in Huh7 cells. **(B)** Quantification of *let-7-a* miRNA levels in hnRNP Q depleted Huh7 cells. **(C)** Northern blot showing increased *let-7-a* levels upon depletion of hnRNP Q in HEK293 cells. **(D)** Quantification of *let-7-a* miRNA levels in hnRNP Q depleted HEK293 cells. **(E)** Northern blot showing no significant change in *miR-21* levels upon depletion of hnRNP Q in Huh7 cells. **(F)** Quantification of *miR-21* miRNA level in hnRNP Q depleted Huh7 cells. Three independent biological replicates were performed.

One representative blot is shown here. The other replicates are shown in **S5 Fig**. Results are plotted as average ± S.D., *$P < 0.05$, **$P < 0.01$, ***$P < 0.001$ using an unpaired two-tailed Student's *t*-test.

TCGA database (**Fig 5E**), which indicates that hnRNP Q is a a prognostic marker for poor survival in liver cancer patients.

## Discussion

Our findings indicate that hnRNP Q/SYNCRIP interacts with LIN28B in an RNA-dependent manner and plays a role in the LIN28/*let-7* regulatory axis. Suppression of hnRNP Q leads to an overall increase in the expression of *let-7* family members and a decrease in levels of the *let-7* target TRIM71. Expression of LIN28, which is also a *let-7* target, is reduced as well. Other RBPs, e.g. hnRNP A and KSRP, have been shown to bind to the terminal loop of pri-*let-7* and regulate the processing of pri- to pre-miRNA by the Microprocessor [48]. A previous study indicated that hnRNP Q can bind pre-*let-7a-2* [49]. Also, suppression of hnRNP Q was reported to impact levels of a number of miRNAs, including *let-7g*, in hepatocellular carcincoma invasive cells [40]. Recently, hnRNP Q was reported to modulate pri-*let-7a* processing in HEK293 cells by binding to its terminal loop. However, that report also showed that suppression of hnRNP Q reduced mature *let-7* levels in HEK293 cells, which was in contrast to our observation [45]. Taken together, we propose that hnRNP Q may regulate *let-7* biogenesis and promote homeostasis through its association and perhaps cooperation with LIN28B in Huh7 cells.

To date, only a fewreports have identified functional partners for LIN28 outside of miRNA biogenesis. A protein-protein interaction between Lin28 and RNA helicase A (RHA) was found to regulate Oct4 mRNAs, likely by influencing translation or mRNA stability [34]. In addition, a mass spectrometry analysis showed that mouse Lin28-containing complexes in muscle cells include several mRNA-binding proteins such as poly(A)-binding protein (PABP), 5′ cap-binding protein, nucleolin, hnRNP F, H1, and insulin-like growth factor (IGF) 2 mRNA-binding proteins (IGF2BPs) [32]. Another study on breast cancer described LIN28A interactions with hnRNP A1, DDX3, Ku70, and PABPC4 [50]. A recent study in epiblast-like stem cells (EpiLCs) showed that the Ddx3x, Hnrnph1, Hnrnpu and Syncrip (Hnrnpq) are necessary for the binding of Lin28a to the Dnmt3a mRNA [38]. These reports indicate that LIN28 could be a part of different oligomeric machinery associated with diverse partners, which includes hnRNP Q. In mice, cytoplasmic hnRNP Q has been shown to compete with PABP and hence impact the global miRNA-mediated deadenylation and translational repression of target mRNAs [51]. Thus, the possibility that suppression of hnRNP Q could similarly alter *let-7* levels through an indirect mechanism cannot be excluded. For example, hnRNP Q1 may have mRNA targets in the cytoplasm for certain genes that possibly control *let-7*, or hnRNP Q1 could share RNA targets with LIN28. However, we have observed that increasing the nuclear-specific hnRNP Q2 and Q3 isoforms also restores TRIM71 and LIN28 expression. Thus, hnRNP Qs are likely at least in part involved in *let-7* biogenesis in the nucleus. Since both LIN28 and TRIM71 form negative feedback controls with *let-7*, depletion of hnRNP Q may also initiate or maintain a cellular environment in which *let-7* level are higher than TRIM71 and LIN28.

LIN28 is a well established stem cell factor, and since its expression is often reactivated in cancer cells, it is also regarded as a pro-oncogene or oncogene. Its oncogenic effect has been shown to be *let-7* dependent while *let-7* directly represses several well-known oncogenes such as RAS, MYC, HMGA2, among others [18]. Comparing this to the oncogenic potential of hnRNP Q is intriguing. hnRNP Q has been implicated in translational control in cancers.

A

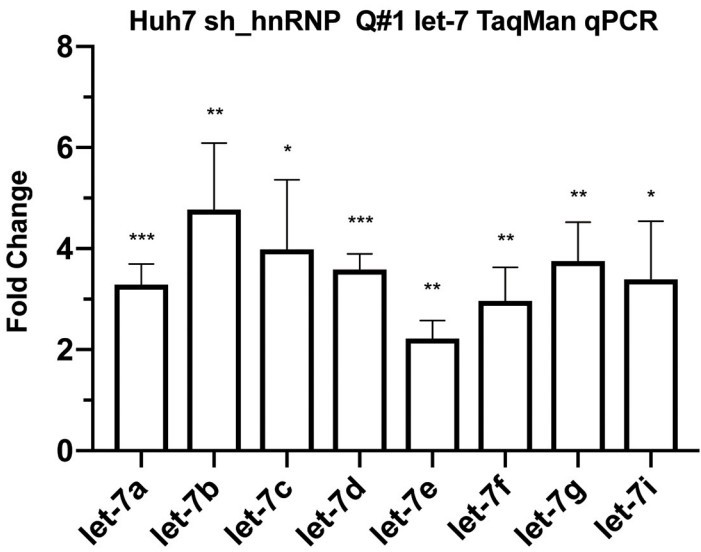

B

Huh7

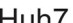
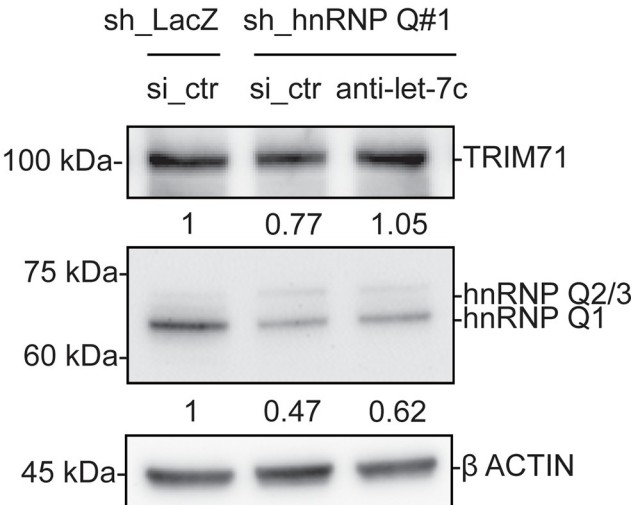

**Fig 4. hnRNP Q depletion increased expression of the *let-7* family miRNAs in Huh7 cells. (A)** The levels of each indicated *let-7* family miRNAs were determined by TaqMan qRT-PCR. Data were shown as foldchange (control = 1) ± S.D. from three independent experiments, *$P < 0.05$, **$P < 0.01$, ***$P < 0.001$ using an unpaired two-tailed Student's *t*-test. U6 snRNA was used as an endogenous control. **(B)** Immunoblot showing transfection of anti-*let-7c* restored the level of TRIM71 in Huh7 cells when hnRNP Q was knocked down.

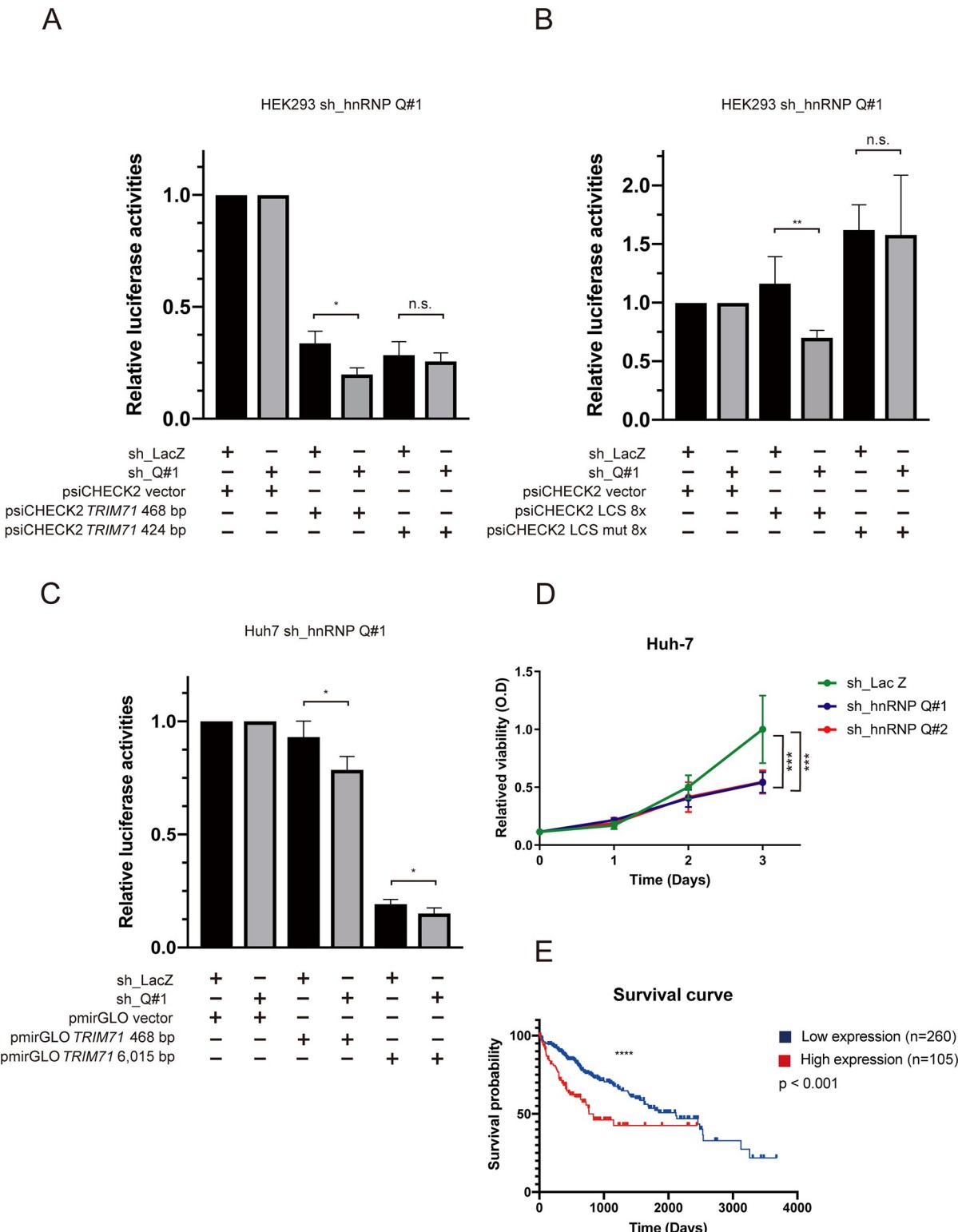

**Fig 5. hnRNP Q depletion reduced the relative luciferase activities of *let-7* complementary sequence (LCS)-bearing luciferase in HEK293 and Huh7 cells.** (**A**) hnRNP Q depletion significantly reduced the relative luciferase activities of Trim71 3'UTR 468-bearing luciferase while having no effect on luciferase-containing 3'UTRs without a LCS. The psiCHECK2 vector carrying either the 468 bp TRIM71 3'UTR or the 424-bp 3'UTR with LCSs-deleted were transfected into control knockdown (black bars) or hnRNP Q knockdown (grey bars) HEK293 cells. For both the hnRNP Q knocked-down or control studies, luciferase activities were normalized to the samples with the psiCHECK2 empty

vector. **(B)** hnRNP Q depletion significantly reduced relative luciferase activities of LCS-bearing luciferase in HEK293 cells. The psiCHECK2 vectors carrying 8X LCSs or 8X mutated LCSs were transfected into hnRNP Q knockdown (in grey) or control knockdown (in black) HEK293 cells for the assay. **(C)** hnRNP Q depletion significantly reduced the relative activities for luciferase constructs carrying either the 468 bp or 6,015 bp Trim71 3'UTRs in Huh7 cells. The pmirGLO vector containing either the 468 bp or 6,015 bp TRIM71 3'UTR were transfected into control knockdown (black bars) or hnRNP Q knockdown (grey bars) Huh7 cells for the assay. Results were plotted as foldchange (control = 1 for control knockdown or hnRNP Q knockdown samples for each set of luciferase activity tests) ± S.D. **(D)** Cell proliferation of Huh7 cells infected with lentiviral constructs containing sh_LacZ, sh_ hnRNP Q#1, or sh_ hnRNP Q#2 was determined by the MTT assay at the indicated time points. For Fig 5A and 5C, N = 4; Fig 5B and 5D, N = 3. n.s., not significant. *$P < 0.05$, **$P < 0.01$, ***$P < 0.001$ using an unpaired two-tailed Student's $t$-test. **(E)** Mantel-Cox survival curves comparing the effects of hnRNP Q expression in liver cancer.

Overexpression of hnRNP Q1 positively correlates with overexpressed Aurora-A in clinical colorectal cancer tissues, where it binds to the Aurora-A 5'-UTR and regulates translation [41]. In myeloid leukemia, hnRNP Q upregulates mRNA targets, including the critical genes Myc, Hoxa9, and Ikzf2, and promotes the stem cell program [39]. Activation of Myc has been shown to repress widespread miRNA expression, including *let-7* [52]. Myc-mediated *let-7* repression is through the activation of LIN28B promoter activity by Myc [53]. Moreover, our study showed that hnRNP Q depletion inhibited the proliferation of Huh7 cells, which is consistent with the results in the TCGA database that a high level of hnRNP Q is considered an unfavorable prognostic marker for liver cancer. Collectively, although other factors may still be involved in the hnRNP Q modulates LIN28B/let-7 axis, our results suggest that hnRNP Q, LIN28, and let-7 interact in carcinogenesis.

In conclusion, our studies demonstrate that hnRNP Q is a LIN28-interacting protein that is required for a low expression level state for *let-7* in Huh7 cells. This adds a new layer of regulation to the LIN-28/*let-7* axis and provides a new approach for studying miRNA-related diseases.

## Supporting information

**S1 Fig. (A)** Graphic depiction of the hnRNP Q gene. Three major isoforms are caused by alternative splicing. The longest isoform hnRNP Q3 represents most of the RGG box at the C-terminal. hnRNP Q2 contains a truncated second RBD due to alternative splicing at exon 7. hnRNP Q1 contains a truncated C-terminal due to alternative splicing. **(B)** Graphical depiction of the pEF-hnRNP Q1/2/3 constructs. **(C)** Design of the RNAi-resistant sequence for hnRNP Q.
(TIF)

**S2 Fig. hnRNP Q depletion by shRNAs reduces protein levels of TRIM71 and LIN28B in Huh7 cells.** Three other biological replicates that accompany Fig 2A. The protein levels of hnRNP Q, TRIM71 and LIN28 were analyzed by western blot. Actin served as an internal control.
(TIF)

**S3 Fig. hnRNP Q depletion by siRNAs reduces protein levels of TRIM71 and LIN28B in Huh7 cells.** Huh7 cells were transfected with three different siRNAs against hnRNP Q. The protein level of hnRNP Q, TRIM71 and LIN28 were analyzed by western blot. Actin served as an internal control. **(A)** Two other biological replicates for hnRNP Q depletion by shRNAs reduces protein levels of TRIM71 and LIN28B in Huh7 cells. **(B)** Quantification of hnRNP Q levels in Huh7 cells with siRNA. **(C)** Quantification of TRIM71 levels in hnRNP Q depleted Huh7 cells. **(D)** Quantification of LIN28B levels in hnRNP Q depleted Huh7 cells. Results are plotted as average ± S.D., *$P < 0.05$, **$P < 0.01$, ***$P < 0.001$ using an unpaired two-tailed Student's $t$-test.
(TIF)

**S4 Fig. Determination of the intracellular distribution of endogenous LIN28B, hnRNP Q isoforms, and HA-tagged hnRNP Q isomers in Huh7 cells. (A)** Nuclear/cytoplasmic fractionation of Huh7 cells without or with transfected HA-tagged hnRNP Q isoforms. The distribution of hnRNP Q isoforms for both the wild-type hnRNP Q and HA-tagged hnRNP Q was detected using an anti-hnRNP Q antibody. hnRNP Q1 predominantly localized to the cytoplasm while the vast majority of hnRNP Q2 and 3 was found in the nucleus. We found that LIN28B was detected in both the nucleus and cytoplasm. LAMIN A/C and GAPDH served as nuclear and cytoplasmic markers, respectively **(B)** The HA-tagged hnRNP Q isoforms were detected by western blot using an anti-HA antibody. The localization of HA-tagged hnRNP Q isoforms was the same as observed with endogenous proteins. **(C)** Immunocytochemistry of endogenous hnRNP Q and LIN28B in Huh7 cells. hnRNP Q (labeled in green) was found in both the cytoplasm and the nucleus while LIN28B (labeled in red) localized to the cytoplasm and the nucleolus. DAPI staining indicated the nucleus. Three biological replicates are shown. **(D)** Immunocytochemistry of HA-tagged hnRNP Q isoforms and endogenous Lin28B in Huh7 cells. HA-tagged hnRNP Q1 localized to the cytoplasm and the nucleus while hnRNP Q2 and 3 predominantly localized to the nucleus.
(TIF)

**S5 Fig. Two other biological replicates for *let-7-a* and *miR-21* expression shown in Fig 3.**
(TIF)

**S6 Fig. The levels of each indicated *let-7* family of miRNAs were determined by TaqMan qRT-PCR.** Data were shown as foldchange (Ctr = 1) ± S.D. from three independent experiments, $*P < 0.05$, $**P < 0.01$, $***P < 0.001$ using an unpaired two-tailed Student's $t$-test. RNU48 snoRNA was used as an endogenous control.
(TIF)

**S7 Fig. Schematic of the luciferase assay vectors used in this work. (A)** The psiCHECK2 vector. **(B)** A 468-bp TRIM71 3'UTR and a 424-bp TRIM71 3' UTR with two LCSs deleted were cloned into the psiCHECK2 downstream of the renilla luciferase gene. **(C)** 3'UTR sequences containing eight copies of LCSs or mutated LCSs were cloned into the psiCHECK2 vector. **(D)** The pmirGLO vector. **(E)** A 468-bp TRIM71 3'UTR and a 6015-bp TRIM71 3' UTR were cloned into the pmirGLO after the firefly luciferase gene.
(TIF)

**S1 Table. The primers used in this work.**
(PDF)

**S1 Raw images. Original western blot and northern blot images.**
(PDF)

## Acknowledgments

We express gratitude to Dr. Tarn (IBMS, Academia Sinica, Taiwan) for generously providing hnRNP Q 2 and 3 isoforms. Special thanks to James Ou (University of California, U.S) and Dr. Miniyi Liu (Department of Biochemistry and Molecular Biology, National Taiwan University College of Medicine, Taiwan) for sharing the Huh7 cell line. We also appreciate to Dr. Yan (Department of Biochemistry and Molecular Biology, National Taiwan University College of Medicine, Taiwan) for technical assistance. Our thanks extend to the Microbiology Institute and the First Core Lab at the National Taiwan University College of Medicine for their valuable services.

## Author Contributions

**Conceptualization:** Jason Jei-Sheng Chang, Shih-Peng Chan.

**Data curation:** Jason Jei-Sheng Chang, Ti Lin, Xin-Yue Jhang.

**Formal analysis:** Jason Jei-Sheng Chang, Ti Lin, Xin-Yue Jhang.

**Funding acquisition:** Shih-Peng Chan.

**Investigation:** Jason Jei-Sheng Chang, Ti Lin, Xin-Yue Jhang.

**Methodology:** Jason Jei-Sheng Chang, Ti Lin, Xin-Yue Jhang.

**Project administration:** Jason Jei-Sheng Chang, Ti Lin, Shih-Peng Chan.

**Resources:** Jason Jei-Sheng Chang, Ti Lin, Xin-Yue Jhang.

**Supervision:** Shih-Peng Chan.

**Validation:** Jason Jei-Sheng Chang, Ti Lin, Xin-Yue Jhang.

**Visualization:** Jason Jei-Sheng Chang, Ti Lin.

**Writing – original draft:** Jason Jei-Sheng Chang, Ti Lin, Shih-Peng Chan.

**Writing – review & editing:** Jason Jei-Sheng Chang, Ti Lin, Shih-Peng Chan.

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
