## [Decision Letter · Decision Letter 0]

27 Dec 2023

PONE-D-23-40197hnRNP Q/SYNCRIP interacts with LIN28 and modulates the LIN28/let-7 axis in human hepatoma cellsPLOS ONE

Dear Dr. Chan,

Thank you for submitting your manuscript to PLOS ONE. After careful consideration, we feel that it has merit but does not fully meet PLOS ONE’s publication criteria as it currently stands. Therefore, we invite you to submit a revised version of the manuscript that addresses the points raised during the review process.More specifically, modify the text and figures following the suggestions of reviewer 1 and 2.

We look forward to receiving your revised manuscript.

Kind regards,

Massimo Caputi, PhD

Academic Editor

PLOS ONE

Journal Requirements:

3. Thank you for stating the following financial disclosure: "1.Ministry of Science and Technology, Taiwan (MOST 105-2311-B-002-011-MY3) 

2.Ministry of Science and Technology, Taiwan (MOST 108-2311-B-002-012) 

3.Ministry of Science and Technology, Taiwan (MOST 111-2314-B002-065-MY3)

4.National Taiwan University Hospital (UN108-040)

5.National Taiwan University (109L7224)

6.National Taiwan University (110L7205)

7.National Taiwan University Hospital Yunlin Branch (NTUHYL110.I003)

8.National Taiwan University Hospital Yunlin Branch (NTUHYL111.I009)"

4. Thank you for stating the following in the Acknowledgments Section of your manuscript: "We express gratitude to Dr. Tarn (IBMS, Academia Sinica, Taiwan) for generously providing hnRNP Q 2 and 3 isoforms. Special thanks to James Ou (University of California, U.S) and Dr. Miniyi Liu (Department of Biochemistry and Molecular Biology, National Taiwan University College of Medicine, Taiwan) for sharing the Huh7 cell line. We also appreciate to Dr. Yan (Department of Biochemistry and Molecular Biology, National Taiwan University College of Medicine, Taiwan) for technical assistance. Our thanks extend to the Microbiology Institute and the First Core Lab at the National Taiwan University College of Medicine for their valuable services. This work was supported by grants from the Ministry of Science and Technology, Taiwan (MOST 105-2311-B-002-011-MY3, 108-2311-B-002-012, 111-2314-B002-065-MY3), National Taiwan University Hospital (UN108-040), National Taiwan University (109L7224 and 110L7205), and National Taiwan University Hospital Yunlin Branch (Z110.I003 and 111.I009) to SPC."

Please remove any funding-related text from the manuscript and let us know how you would like to update your Funding Statement. Currently, your Funding Statement reads as follows: "1.Ministry of Science and Technology, Taiwan (MOST 105-2311-B-002-011-MY3) 

2.Ministry of Science and Technology, Taiwan (MOST 108-2311-B-002-012) 

3.Ministry of Science and Technology, Taiwan (MOST 111-2314-B002-065-MY3)

4.National Taiwan University Hospital (UN108-040)

5.National Taiwan University (109L7224)

6.National Taiwan University (110L7205)

7.National Taiwan University Hospital Yunlin Branch (NTUHYL110.I003)

8.National Taiwan University Hospital Yunlin Branch (NTUHYL111.I009)"

5. We note that your Data Availability Statement is currently as follows: "All relevant data are within the manuscript and its Supporting Information files."

Reviewers' comments:

Reviewer's Responses to Questions

**Comments to the Author**

1. Is the manuscript technically sound, and do the data support the conclusions?

Reviewer #1: Yes

Reviewer #2: Yes

2. Has the statistical analysis been performed appropriately and rigorously? 

Reviewer #1: No

Reviewer #2: Yes

3. Have the authors made all data underlying the findings in their manuscript fully available?

Reviewer #1: Yes

Reviewer #2: Yes

4. Is the manuscript presented in an intelligible fashion and written in standard English?

Reviewer #1: Yes

Reviewer #2: Yes

5. Review Comments to the Author

Reviewer #1: This is an interesting study that contains novel results regarding hnRNP Q and the interaction with LIN28.

One of the main findings in the manuscript is that reducing the amount of hnRNP Q resulted in a decrease in LIN28B and TRIM71. These results are shown in figure 2A and supplemental figure 2. I would like to request a little more clarity around these 2 figures. Please include how many reps? What are the quantification numbers? Are the numbers an average or just this rep? Are the reductions significantly different?

The siRNA experiment in supplemental figure 2 did not reduce TRIM71 or LIN28B as much as the shRNA experiments in figure 2a. Since the levels of hnRNP Q were reduced in the si experiments, to about the same levels, I feel like this needs to have a little more explanation than the lines 356-359. A clear explanation of the quantification and statistical analysis may hope to clarify this as well.

Reviewer #2: This is a well written manuscript with a set of thorough experiments with sound conclusions. Just some minor changes that I think should be easy to implement:

1. I think the authors should be careful about making statements like their results are consistent with studies showing LIN28B and TRIM71 promote growth and tumorigenicity when they have not done any experiments to investigate the role of these proteins in growth or tumorigenicity. This is especially misleading in the abstract and needs to be taken out (lines 38-40).

2. For Fig 1 I wonder if all RBPs are sensitive to RNAse A treatment and would show lower amounts in IP. Would be great if they could add another control thats a different RBP. Also what does the asterisk under LIN28B refer to in C/D/E?

3. There are several typos that need to be fixed. Line 24: tumor-suppression miRNA should be 'tumor suppressor'; line 34-35: miRNAs levels should be miRNA levels; line 71: 'oligouridylation following' should be 'oligouridylation followed by'; line 98: the a common G quartet structure. Not sure what the authors are trying to state. There are some others in the manuscript and in the supplementary legends.

4. The authors should try to be specific whenever possible. Instead of stating LIN28 in the abstract, please state LIN28B as the observations in this study relate only to this homolog. Similarly when performing Northern blot, they should state let-7a and not just let-7. In line 488 they say unfavorable prognostic marker for liver cancer- is it really all liver cancer or just HCC- please be specific. In figure legend 5e (Line 569) they again say liver cancer.

5. Please provide as much detail as possible in the figure legends- for e.g. in the legend for Fig 2 they do not explain Alpha-HA. I understand they are referring to antibody to the HA tag but for someone who has not had the chance to read the text, it is not clear.

6. Figure 3E legend (line 531) incorrectly states blot showing increased miR-21 levels upon depletion... what they mean to say is unchanged miR-21 levels.

7. While the authors have established a possible mechanism of how hnRNPQ is regulating LIN28B through let-7, its also possible that there may be other possible explanations- for e.g. LIN28B and hnRNPQ may be both regulated by some common factor that promotes their stability when they come together through their interaction with some common RNA molecule. The authors (to their credit) speculate that hnRNPQ1 could share RNA targets with LIN28B, they then state that since increasing Q2 and Q3 isoforms also restores TRIM71 and LIN28B it is likely that hnRNPQs are involved with let-7 biogenesis. It does seem to be the most logical explanation but other possible mechanisms (such as hnRNPQ regulating LIN28 through MYC) may also be at play.

6. PLOS authors have the option to publish the peer review history of their article (what does this mean?). If published, this will include your full peer review and any attached files.

Reviewer #1: No

Reviewer #2: **Yes: **Shubin Shahab

---

## [Author Response · Author response to Decision Letter 0]

8 Apr 2024

Reviewer #1: This is an interesting study that contains novel results regarding hnRNP Q and the interaction with LIN28. One of the main findings in the manuscript is that reducing the amount of hnRNP Q resulted in a decrease in LIN28B and TRIM71. These results are shown in figure 2A and supplemental figure 2. I would like to request a little more clarity around these 2 figures. Please include how many reps? What are the quantification numbers? Are the numbers an average or just this rep? Are the reductions significantly different?

We thank the reviewer for the thorough comments. In the original Figure 2A and Supplementary Figure 2, we only presented one western blot result and included a single quantitative value. Now, we provide additional western blot results with shRNA knockdown (Supplementary Fig. S2A-C) or siRNA knockdown (Supplementary Fig. S3B-C). We tested whether hnRNP Q depletion reduced the protein levels of TRIM71 and LIN28B (Fig. 2B-D and Supplementary Fig. S3B-D) using unpaired Student's t-tests, and the reductions were significantly different. We believe this explanation answers the reviewer's question.

siRNA

hnRNP Q

si_hnRNP Q#1 vs. si_Ctr

P value: 0.0140

si_hnRNP Q#2 vs. si_Ctr

P value: 0.0264

si_hnRNP Q#3 vs. si_Ctr

P value: 0.0103

TRIM71

si_hnRNP Q#1 vs. si_Ctr

P value: 0.0395

si_hnRNP Q#2 vs. si_Ctr

P value: 0.0105

si_hnRNP Q#3 vs. si_Ctr

P value: 0.0214

LIN28B

si_hnRNP Q#1 vs. si_Ctr

P value: 0.0081

si_hnRNP Q#2 vs. si_Ctr

P value: 0.0452

si_hnRNP Q#3 vs. si_Ctr

P value: 0.0357

shRNA

hnRNP Q

sh_hnRNP Q#1 vs. sh_LacZ

P value: 0.0011

sh_hnRNP Q#2 vs. sh_LacZ

P value: 0.0105

TRIM71

sh_hnRNP Q#1 vs. sh_LacZ

P value: 0.0110

sh_hnRNP Q#2 vs. sh_LacZ

P value: 0.0044

LIN28B

sh_hnRNP Q#1 vs. sh_LacZ

P value: 0.0023

sh_hnRNP Q#2 vs. sh_LacZ

P value: 0.0489

The siRNA experiment in supplemental figure 2 did not reduce TRIM71 or LIN28B as much as the shRNA experiments in figure 2a. Since the levels of hnRNP Q were reduced in the si experiments, to about the same levels, I feel like this needs to have a little more explanation than the lines 356-359. A clear explanation of the quantification and statistical analysis may hope to clarify this as well.

We thank the reviewer for their insightful comments. The electrochemiluminescence (ECL) images were measured by the UVP reader and then the intensities of the proteins were quantified using the ImageQuant 800 software, with beta actin as an internal control. The statistical results showed that both shRNA and siRNA significantly knocked down hnRNP Q to similar levels. However, during siRNA transfection, there was no puromycin selection process like in the shRNA experiments, perhaps leading to less efficacy on changing TRIM71 and LIN28B levels. For example, perhaps the majority of the cells in the shRNA experiment had similar levels of knockdown of hnRNP Q, but in the siRNA experiments, some cells were strongly knocked down while other cells had no knockdown. Therefore, we changed the description on lines 371-373:

[old] “A similar pattern was seen when we used exogenous siRNAs to reduce hnRNP Q expression (Supplementary Fig. S3A), although decreases in LIN28B and TRIM71 levels were more subtle, perhaps due to less efficient knockdown.”

To:

[new] "A similar pattern was seen when we used exogenous siRNA to reduce hnRNP Q expression (Supplementary Figure S3A-D), although decreases in LIN28B and TRIM71 levels were more subtle. One possible explanation for the smaller degree of decrease in the siRNA experiments could be due to differences in the assays; the shRNA experiments used stable clones while the siRNA transfection experiments did not. Reviewer #2: This is a well written manuscript with a set of thorough experiments with sound conclusions. Just some minor changes that I think should be easy to implement: 1. I think the authors should be careful about making statements like their results are consistent with studies showing LIN28B and TRIM71 promote growth and tumorigenicity when they have not done any experiments to investigate the role of

these proteins in growth or tumorigenicity. This is especially misleading in the abstract and needs to be taken out (lines 38-40). We thank the reviewer for the comments. As suggested, we have deleted this description (on lines 38-40) about previous studies showing that LIN28B and TRIM71 promote growth and tumorigenicity. 2. For Fig 1 I wonder if all RBPs are sensitive to RNAse A treatment and would show lower amounts in IP. Would be great if they could add another control thats a different RBP. Also what does the asterisk under LIN28B refer to in C/D/E? The reviewer makes an interesting point that we think would be intriguing to pursue in future studies. There certainly may be other RBPs that are sensitive to RNAse A treatment and then show lower amounts of IP. If there are many, then this may imply that LIN28B interacts with RBPs in a generalist fashion. Alternatively, there may be only one or a few specific RBPs that interact with LIN28B. Both would be interesting to uncover. However, we feel the topic is outside of the scope of the manuscript. We already qualified our conclusions and hinted at this possibility in our prior (and current) draft of the manuscript (copied below) and we prefer to stay focused on the hnRNP Q interaction with LIN28B and leave it here. [prior and current version; highlighted in blue background for emphasis] We detected RNA-dependent interactions between all three major HA-tagged hnRNP Q isoforms and LIN28B (Fig. 1C-E), indicating that despite differences in structure and localization, the hnRNP Q isoforms may share RNA targets with LIN28 or precipitate into the same ribonucleotide protein particles within LIN28B. The "*" represents a non-specific band, and we have added this description to the figure 1 legend. 3. There are several typos that need to be fixed. Line 24: tumor-suppression miRNA should be 'tumor suppressor'; line 34-35: miRNAs levels should be miRNA levels;

line 71: 'oligouridylation following' should be 'oligouridylation followed by'; line 98: the a common G quartet structure. Not sure what the authors are trying to state. There are some others in the manuscript and in the supplementary legends. Thank for catching these typos. We have made the changes accordingly, including line 24: “tumor-suppression miRNA” to “tumor suppressor”, line 34-35: “miRNAs levels” to “miRNA levels”, and line 72: “oligouridylation following” to “oligouridylation followed by”. Other typos can be seen in the tracked-changes version. We thank the reviewer for the insightful comments. Previous studies pointed out many LIN28 mRNAs targets contain G-quartet (G4) structures. However, our results do not provide information about how RNA is bound to LIN28B. Therefore, we decided to delete the description (in line 97) “the a common G-quartet structure" from the article. 4. The authors should try to be specific whenever possible. Instead of stating LIN28 in the abstract, please state LIN28B as the observations in this study relate only to this homolog. Similarly when performing Northern blot, they should state let-7a and not just let-7. In line 488 they say unfavorable prognostic marker for liver cancer- is it really all liver cancer or just HCC- please be specific. In figure legend 5e (Line 569) they again say liver cancer. Thank for your corrections. We have changed the description of “LIN28” in the abstract for more specific to “LIN28B”. We have changed “let-7” in the description of northern blot to “let-7-a”. According to the TCGA database, high levels of hnRNP Q are considered to be an unfavorable prognostic marker, which is statistically associated with liver cancer (Hepatocellular carcinoma (HCC), Intrahepatic cholangiocarcinoma, Angiosarcoma and hemangiosarcoma, and Hepatoblastoma) patients. Also, Llovet, J.M., Kelley, R.K., Villanueva, A. et al (2021) reported that HCC is the most common form of liver cancer and accounts for ~90% of cases. Here, we quoted the results of the TCGA database in line 488 and line 569, we describe it as “liver cancer” instead of “HCC”.

5. Please provide as much detail as possible in the figure legends- for e.g. in the legend for Fig 2 they do not explain Alpha-HA. I understand they are referring to antibody to the HA tag but for someone who has not had the chance to read the text, it is not clear. We thank the reviewer for the thorough suggestion. We have changed the α-HA in Figure 2E to anti-HA. 6. Figure 3E legend (line 531) incorrectly states blot showing increased miR-21 levels upon depletion... what they mean to say is unchanged miR-21 levels. Thank for your correction. We have made the changes accordingly on line 531 and changed "Northern blot showing increased in miR-21 levels upon depletion of hnRNP Q in Huh7 cells." to "Northern blot showing no significant change in miR-21 levels upon depletion of hnRNP Q in Huh7 cells." 7. While the authors have established a possible mechanism of how hnRNPQ is regulating LIN28B through let-7, its also possible that there may be other possible explanations- for e.g. LIN28B and hnRNPQ may be both regulated by some common factor that promotes their stability when they come together through their interaction with some common RNA molecule. The authors (to their credit) speculate that hnRNPQ1 could share RNA targets with LIN28B, they then state that since increasing Q2 and Q3 isoforms also restores TRIM71 and LIN28B it is likely that hnRNPQs are involved with let-7 biogenesis. It does seem to be the most logical explanation but other possible mechanisms (such as hnRNPQ regulating LIN28 through MYC) may also be at play.

We appreciate this comment and agree with the reviewer that there may be other players involved in the hnRNP Q modulation of LIN28. We have already discussed the possibility of other players on lines 548-559: [Its oncogenic effect has been shown to be let-7 dependent while let-7 directly represses several well-known oncogenes such as RAS, MYC, HMGA2, among others [19]. Comparing this to the oncogenic potential of hnRNP Q is intriguing. hnRNP Q has been implicated in translational control in cancers. Overexpression of hnRNP Q1 positively correlates with overexpressed

Aurora-A in clinical colorectal cancer tissues, where it binds to the Aurora-A 5'-UTR and regulates translation [43]. In myeloid leukemia, hnRNP Q upregulates mRNA targets, including the critical genes Myc, Hoxa9, and Ikzf2, and promotes the stem cell program [41]. Activation of Myc has been shown to repress widespread miRNA expression, including let-7 [54]. Myc-mediated let-7 repression is through the activation of LIN28B promoter activity by Myc [55].] However, whether in HCC hnRNPQ also upregulates MYC levels is unclear and awaiting to be answered. Therefore, we have now added a qualifying sentence (on lines 496-497): Collectively, although other factors may still be involved in the hnRNP Q modulates LIN28B/let-7 axis, our results suggest that hnRNP Q, LIN28, and let-7 interact in carcinogenesis. Reference Llovet, J.M., Kelley, R.K., Villanueva, A. et al. Hepatocellular carcinoma. Nat Rev Dis Primers 7, 6 (2021). https://doi.org/10.1038/s41572-020-00240-3

---

## [Decision Letter · Decision Letter 1]

22 May 2024

hnRNP Q/SYNCRIP interacts with LIN28B and modulates the LIN28B/let-7 axis in human hepatoma cells

PONE-D-23-40197R1

Dear Dr. Chan,

We’re pleased to inform you that your manuscript has been judged scientifically suitable for publication and will be formally accepted for publication once it meets all outstanding technical requirements.

Kind regards,

Massimo Caputi, PhD

Academic Editor

PLOS ONE

Additional Editor Comments (optional):

Reviewers' comments:

Reviewer's Responses to Questions

**Comments to the Author**

1. If the authors have adequately addressed your comments raised in a previous round of review and you feel that this manuscript is now acceptable for publication, you may indicate that here to bypass the “Comments to the Author” section, enter your conflict of interest statement in the “Confidential to Editor” section, and submit your "Accept" recommendation.

Reviewer #1: All comments have been addressed

Reviewer #2: All comments have been addressed

2. Is the manuscript technically sound, and do the data support the conclusions?

Reviewer #1: (No Response)

Reviewer #2: Yes

3. Has the statistical analysis been performed appropriately and rigorously? 

Reviewer #1: (No Response)

Reviewer #2: Yes

4. Have the authors made all data underlying the findings in their manuscript fully available?

Reviewer #1: (No Response)

Reviewer #2: Yes

5. Is the manuscript presented in an intelligible fashion and written in standard English?

Reviewer #1: (No Response)

Reviewer #2: Yes

6. Review Comments to the Author

Reviewer #1: (No Response)

Reviewer #2: Thank you for making the changes. Few more minor suggestions:

Minor grammatical error in lines 562-564. Perhaps say :"Collectively, although other factors may still be involved in how hnRNP Q modulates the LIN28B/let-7 axis, ..."

I would also suggest some minor changes in wording. MTS assay does not necessarily measure proliferation (which is measured by cell cycle entry), but rather "viability". in line 472-473, I would suggesting stating "... increase of let-7 family miRNAs, which in turn would decrease levels of their targets..". Expression implies mRNA transcription. We do not know whether let-7 inhibits transcription/translation or causes degradation of the transcript.

7. PLOS authors have the option to publish the peer review history of their article (what does this mean?). If published, this will include your full peer review and any attached files.

Reviewer #1: No

Reviewer #2: **Yes: **Shubin Shahab

---

## [Editor Report · Acceptance letter]

27 Jun 2024

PONE-D-23-40197R1 

PLOS ONE

Dear Dr. Chan, 

I'm pleased to inform you that your manuscript has been deemed suitable for publication in PLOS ONE. Congratulations! Your manuscript is now being handed over to our production team.

Kind regards, 

on behalf of

Dr. Massimo Caputi 

Academic Editor

PLOS ONE